

**Technical Note: Improved mathematical representation of**
**concentration-discharge relationships**
José Manuel Tunqui Neira[1, 2], Vazken Andréassian[1], Gaëlle Tallec[1] & Jean-Marie Mouchel [2]
[(1)] Irstea, HYCAR Research Unit, Antony, France
[(2)] Sorbonne University, UMR Metis 7619, Paris, France
**Abstract**
This Technical Note deals with the mathematical representation of concentration-discharge
relationships and with the identification of its parameters. We propose a two-sided power
transformation alternative to the classical log-log transformation, and a multicriterion identification
procedure allowing determining parameters that are efficient, both from the concentration and the
load points of view.
**Keywords**
Concentration-discharge relationships; log-log transformation; power transformation; multi-
objective calibration

**1.    Introduction**
The relationship between ion concentrations and river discharge is an age-old topic in hydrology (see
among others Bazerbachi and Probst, 1986;Durum, 1953;Foster, 1978;Gibbs, 1970;Gregory and
Walling, 1973;Hem, 1948;Hendrickson and Krieger, 1960;Johnson et al., 1969;Meybeck, 1976).
Several recent articles (e.g. Bieroza et al., 2018;Chanat et al., 2002;Godsey et al., 2009;Meybeck and
Moatar, 2012;Moatar et al., 2017;Rose et al., 2018;Kirchner, 2019) give an exhaustive view of the
current work on this relationship. Since at least seventy years, a one-sided power relationship has
been used to represent and model this relationship (Eq. (1)).

$$C = aQ^b \qquad \text{Eq. (1)}$$

$$\ln(C) = \ln(a) + b.\ln(Q) \qquad \text{Eq. (2)}$$




From a graphical point of view, the one-sided power relation presented in Eq. (1) is equivalent to
plotting concentration and discharge in a log-log space, where parameters $a$ and $b$ can be identified
very simply, either graphically or numerically (under the classical assumptions of linear regression).
Obviously, not all ion species present a clear relationship between concentration and discharge; but
a few exhibit a very clear one as shown by the Figure 1 which gives an example from Upper Hafren
catchment data (Neal et al., 2013a;Neal et al., 2013b). In the rest of this note, we will focus explicitly
on the ions which exhibit a clear dependency between discharge and concentration.

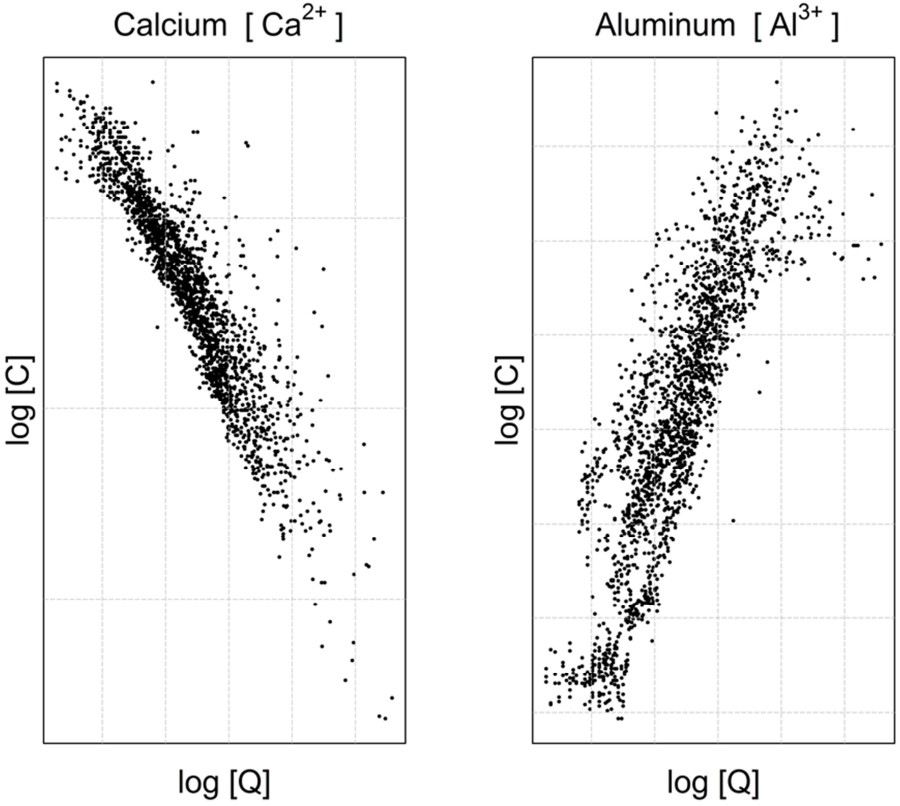


**Figure 1. Illustration of the C-Q behavior of two ions (Calcium and Aluminum) showing a clear log-**
**log relationship. Data from Upper Hafren catchment, Wales, United Kingdom (Neal et al.,**
**2013a;Neal et al., 2013b)**

## 2.    About the "excess" of the log-log transformation
For many years, since the size of the C-Q datasets was limited by the cost of chemical analyses, it was
difficult to analyze in much detail the precise shape of the C-Q relationship. In many cases, the log-



log transformation appeared visually adequate (and conceptually simple), which explains its lasting
popularity. With the advent of high-frequency measuring devices in recent years, the size of the
datasets has exploded, and all the extremes of the relationship can now be included in the analysis.
Figure 2 shows an example from such a high-frequency dataset, collected from the Oracle-Orgeval
observatory (Tallec et al., 2015;Floury et al., 2017). The 17500 data points represent half-hourly
measurements of ions collected over a two-year period, during which the catchment was exposed to
a variety of high- and low-flow events, which provides an unprecedented opportunity for exploring
the shape of the C-Q relationship.

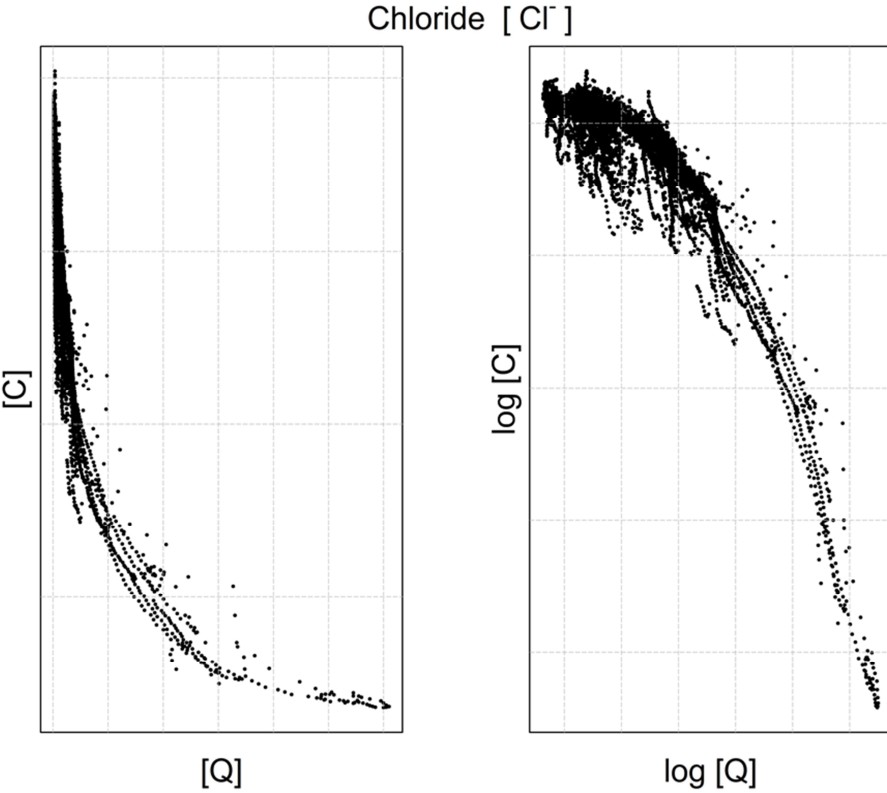


**Figure 2. Concentration-discharge relationship observed on the Oracle-Orgeval observatory**
**(measurements from the "RiverLab") for Chloride ions, with (right) and without (left) logarithm**
**transformation.**

Figure 2 illustrates what we call here the "excess" of the log-log transformation: the C-Q relationship
evolves from a clearly concave shape on the left to a slightly convex shape on the right. We have
gone beyond the straight shape that we aimed at. Note that this is not always the case, and the log-



log transformation can be well adapted in other situations (see Fig. 1 for the Upper Hafren data set
or further examples in the paper by Moatar et al., 2017). This slightly different shape may be due to
the high frequency of the time series (Moatar and Meybeck, 2007) or to catchment dynamics
(Kirchner, 2009), but in any case, it requires a different mathematical treatment than a logarithm
transformation.
**3.   Box-Cox transformations as a continuous alternative to the log-log**
**transformations**
As a progressive alternative to the log-log transformation, we propose to use a two-sided power
transformation as shown in Eq. (3 (Box and Cox, 1964;Howarth and Earle, 1979).

$$C^{\frac{1}{n}} = a + bQ^{\frac{1}{n}}$$    **Eq. (3)**


The advantage of this transformation was already underlined by Box and Cox (1964): when $n$ takes
high values, Eq. (3) converges towards the logarithm transformation (Eq. (2)), thus offering a
progressive solution. The reason is simple:

$C^{\frac{1}{n}} = e^{\frac{1}{n}lnC} \approx 1 + \frac{1}{n}lnC$  when $n$ is large.

Thus for large values of $n$, Eq. (3 can be written:

$$1 + \frac{1}{n}lnC \approx a + b + \frac{b}{n}lnQ$$

That is equivalent to

$lnC \approx A + b.lnQ$ (with $A = n(a + b - 1)$)

The progressive behavior and the convergence towards the log-log transformation are clearly
apparent in Figure 3.

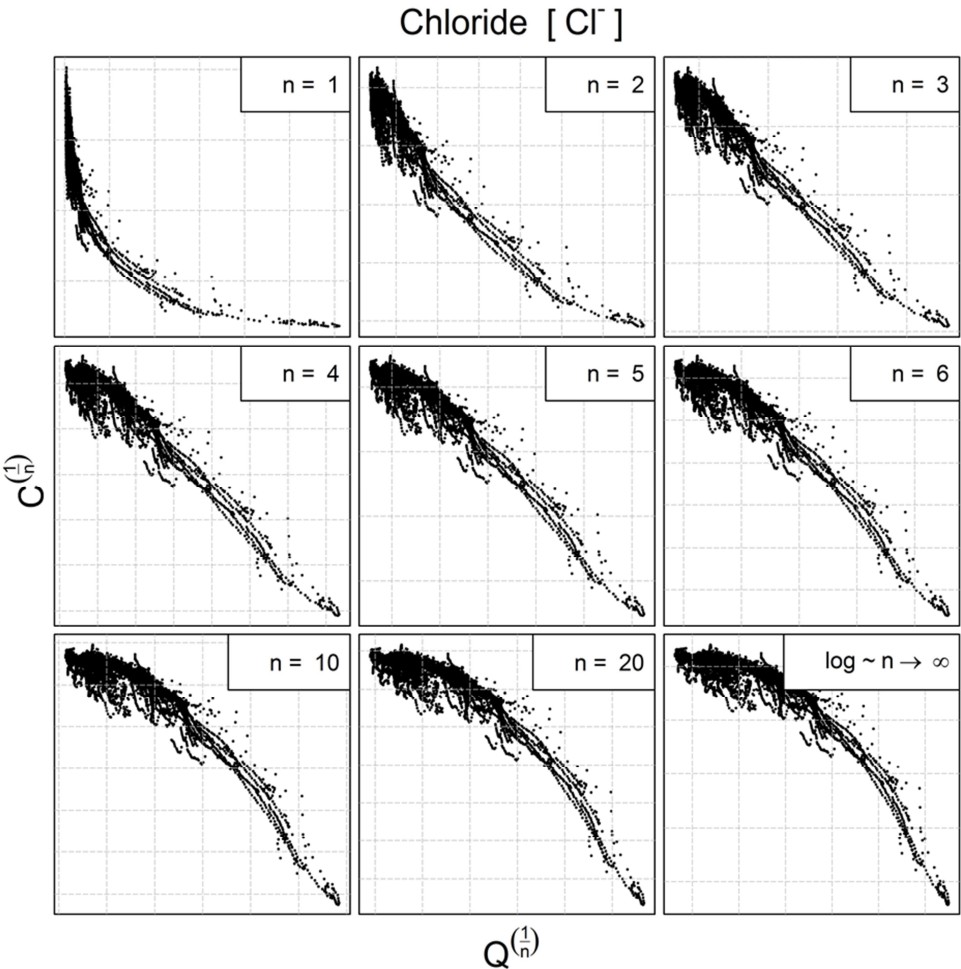


**Figure 3. Evolution of the shape of the concentration-discharge scatterplot with an increasing value**
**of *n*. Chloride ions concentrations measured on the Oracle-Orgeval observatory ("RiverLab").**

## 4. Choosing an appropriate transformation for different ion species

To our knowledge, there is no physical or mathematical reason why all ionic species should have a C-
Q relationship of the same shape. In Figure 4, we show the behavior of 3 ions and EC (Electrical
conductivity) from the same catchment and the same dataset (all four from the Oracle-Orgeval
observatory). The optimal shape could be chosen numerically (see Table 1), but we first followed the
advice of Box et al. (2016, p. 331) and did it visually. Figure 4 shows the most adapted power
transformation selected for each ion and EC.

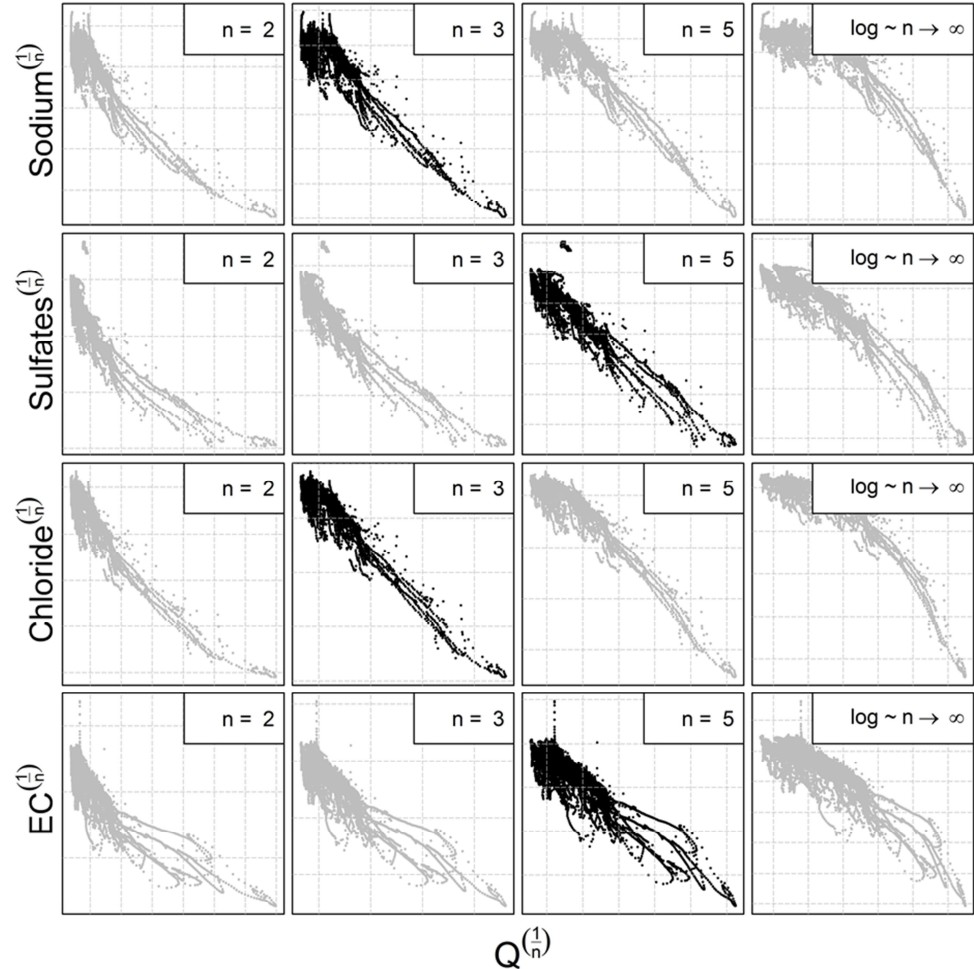


**Figure 4: C-Q behavior of three different chemical species and the conductivity with different**
**transformations (*n* =2, 3, 5 and log). The optimal value of the power transformation (black dots)**
**has been chosen visually.**








**Table 1. Coefficient of determination ($R^2$) calculated for n =1 (no transformation), n = optimal value**
**for two-sided power transformation (Figure 4) and n → ∞ (log-log transformation) for each ion and**
**for EC**

| ion | $n$ | $R^2$ |
|---|---|---|
| Sodium | $n = 1$ (no transformation) | 0.53 |
| | $n = 3$ (optimal) | 0.73 |
| | $n \to \infty$ (log-log) | 0.53 |
| Sulfate | $n = 1$ (no transformation) | 0.32 |
| | $n = 5$ (optimal) | 0.81 |
| | $n \to \infty$ (log-log) | 0.77 |
| Chloride | $n = 1$ (no transformation) | 0.52 |
| | $n = 3$ (optimal) | 0.88 |
| | $n \to \infty$ (log-log) | 0.69 |
| EC | $n = 1$ (no transformation) | 0.38 |
| | $n = 5$ (optimal) | 0.79 |
| | $n \to \infty$ (log-log) | 0.74 |


Although we indicated above that the value of $n$ could be chosen visually, we have also calculated
the coefficient of determination ($R^2$ see Table 1) to confirm our choice numerically. For each ion and
EC, the $n$ considered optimal has the highest $R^2$ value.

## 5. Multi-objective identification of the parameters of the C-Q relationship

Once the most appropriate value of the power transformation has been determined, the numerical
identification of the optimal values of parameters $a$ and $b$ (see. Eq. (3)) should be easy. However, the
extremely large number of values in the high-frequency dataset can prevent a robust identification
over the full range of discharges, because the largest discharge values are in small numbers (in our
dataset only 1% of discharges are in the range [2.6 $m^3s^{-1}$, 12.2 $m^3s^{-1}$], and they correspond to the
lowest concentrations (see Figure 2).
To avoid the difficulties linked with the overrepresentation of low-discharge / high-concentration
data points, we tested successfully a multi-objective criterion for identifying the optimal couple ($a,b$).
We used an optimizing simultaneously on the quality of reproduction of the concentrations and the
load (i.e. the discharge-weighted concentrations); otherwise, the large discharge-low concentration
data points do not have enough weight to influence the selection of the parameter set.
The numerical criterion used for the concentration and the load is a bounded version of the Nash and
Sutcliffe criterion (Mathevet et al., 2006). The Nash and Sutcliffe (1970) criterion (see Eq. (4) and Eq.
(6), in Table 2) is well-known and widely used in the field of hydrology. The rescaling proposed by
Mathevet et al. (2006) transforms NSE in NSEB, which varies between -1 and 1 (see Eq. (5) and Eq. (7)



in Table 2). The advantage of this rescaled version is to avoid the occurrence of large negative values
(the original NSE criterion varies in the range ]-∞, 1]).
Last, we also used a combined criterion for both concentration and load, by averaging $NSEB_{conc}$ and
$NSEB_{load}$ (see Eq. (8) in Table 2).

**Table 2. Numerical criteria used for optimization ($C_{obs}$ – observed concentration, $C_{cal}$ – computed**
**concentration, $Q$ – observed discharge)**

| | |
|---|---|
| $NSE_{conc} = 1 - \dfrac{\sum_t (C_{obs}^t - C_{cal}^t)^2}{\sum_t (C_{obs}^t - \overline{C_{obs}})^2}$ | Eq. (4) |
| $NSEB_{conc} = \dfrac{NSE_{conc}}{2 - NSE_{conc}}$ | Eq. (5) |
| $NSE_{load} = 1 - \dfrac{\sum_t (Q^t C_{obs}^t - Q^t C_{cal}^t)^2}{\sum_t (Q^t C_{obs}^t - \overline{QC_{obs}})^2}$ | Eq. (6) |
| $NSEB_{load} = \dfrac{NSE_{load}}{2 - NSE_{load}}$ | Eq. (7) |
| $NSEB_{comb} = \dfrac{1}{2}(NSEB_{conc} + NSEB_{load})$ | Eq. (8) |


For the three ions and EC of the Oracle-Orgeval observatory, Figure 5 plots the performances of 6725
random pairs of parameters $a$ and $b$ (see Eq. (3)). The Pareto plot allows us to visualize the best
compromise between the criterion focusing on concentration ($NSEB_{conc}$) and the criterion focusing on
load ($NSEB_{load}$). The black point in Figure 5 corresponds to the optimum identified by the combined
criterion $NSEB_{comb}$.

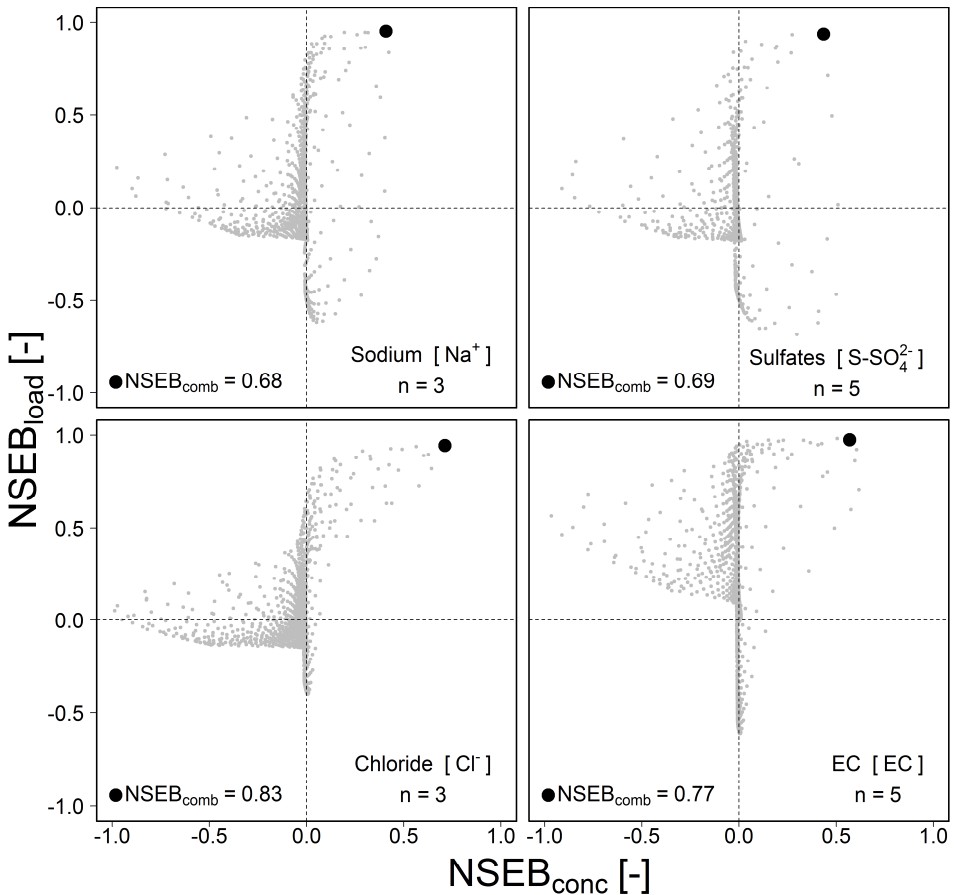


**Figure 5. Multi-objective criterion *NSEB_conc* vs *NSEB_load* (Pareto plot) obtained from 3 chemical species (Sodium, Sulfates, Chloride) and Electrical conductivity (EC) from the Oracle-Orgeval observatory. The black points represent the performance of the pair having the best-combined criterion *NSEB_comb*. Note that like NSE, the optimal value of NSEB is 1, and that a value of 0 shows a very poor fit.**

140

The interest of the Pareto plot is clearly seen in Figure 5. Not all ion species and EC have the same behavior: Chloride has a common optimum for the concentration and the load, while the others do not: they benefit clearly from a compromise that can be either identified visually on the Pareto plot or identified by the combined criterion *NSEB_comb*. The optimal *NSEBc_omb* criterion values for the three





ions and EC range from 0.68 to 0.83 (these values of NSEB correspond to "unbounded" NSE values
ranging from 0.77 to 0.91, which most hydrologists would consider very satisfying).
## 6.    Results
The optimal values of $a$ and $b$ corresponding to the one shown on the Pareto plot (see Figure 5) and
the $n$ value identified  on Figure 4 are presented in Table 3, and Figure 6 illustrates the quality of the
fit over the entire calibration dataset (17500 points). Overall, the two-sided power transformation
model and the multicriterion identification procedure fit very well the concentrations. On can only
mention the difficulty of the model to reproduce the high extreme concentrations and conductivity
that would perhaps require a more elaborated model, which is out of the scope of this Technical
Note.

**Table 3. Summary of values $a$, $b$ and $n$ used to obtain the optimal $NSEB_{comb}$ criterion**

| Ion | $n$ | $a$ | $b$ | $NSEB_{comb}$ |
|---|---|---|---|---|
| **Sodium** | 3 | -0.60 | 2.70 | 0.68 |
| **Sulfate** | 5 | -0.55 | 2.20 | 0.69 |
| **Chloride** | 3 | -1.00 | 3.70 | 0.83 |
| **EC** | 5 | -0.70 | 4.20 | 0.77 |



**Figure 6. Comparison of observed concentrations with simulated concentrations by a two-sided power transformation model:  (a) Scatterplot between the observed and simulated concentration; (b) Comparison of the cumulative frequency of the observed and simulated concentration.**





## 7.    Conclusion

In this technical note, we discussed the log-log transformation, widely used by hydrologists to represent concentration-discharge relationships, and showed that it is sometimes inadequate. The two-sided power transformation we proposed is a valid and progressive alternative. We also showed how the identification of the parameters of this relation can benefit of a multicriterion identification procedure, combining efficiency in concentration and load representation. The simulated concentrations for the 3 ions and the EC show a good performance.

*Data availability*. Data will be available in a dedicated database website after a contract accepted on behalf of all institutes.

*Competing interests*. The authors declare that they have no conflict of interest.

*Acknowledgements.* The first author acknowledges the Peruvian Scholarship Cienciactiva of CONCYTEC for supporting his Ph.D. Study at Irstea and the Sorbonne University. The authors acknowledge the EQUIPEX CRITEX program (grant no. ANR-11-EQPX-0011) for the data availability.

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
