# Peer review of "Technical Note: Improved mathematical representation of"

_Hydrology and Earth System Sciences, 2019_

## Referee Comment (RC1) · Anonymous Referee #1 · 27 Aug 2019

The paper deals with the mathematical representation of the empirical relationships between discharge and ions concentration, presenting an application of the Box and Cox transformation of the data as an alternative to the commonly used log-log transformation. Although the topic is relevant to the current literature and well within the scope of the journal, I am struggling to understand what the exact nature of the problem is, and how and why the proposed work represents an improvement of the current knowledge about the research topic. The scientific significance and quality of the manuscript is quite poor. The literature review presented in the introduction section (rather rushed) does not at all bring the reader to the idea that a different data transformation, beyond the log-log transformation, is desirable by the scientific community for the representa-

tion of Q-C relationship and for what reason it should be. The motivation provided in the "About the excess of log-log transformation" section (the change of the shape of the Q-C relation) is quite weak. Authors probably reach an interesting point/motivation when they introduce (line 111) the problem of the representation of high flow discharge data concentration that arises for high frequency database but, surprisingly, when they come to the results, they mention the difficulties of the model to reproduce this type of data (line 151). But many more questions come about the scientific idea. Why authors choose the Box-Cox transformation? Aren't there alternative? If they do not compare the performance of the log-log transformation with the proposed two-sided power transformation, how can the reader guess it is an improved representation? How is the improvement demonstrated by the authors? The presentation quality is also quite poor. 1) The introduction section is quite rushed and presents a figure published elsewhere by other authors. Generally figures are not included in the introduction but if needed why do not use authors own data? 2) Figures frequently do not indicate neither the range of variability of the data not the unit of measurements 3) The dataset used for the analysis is not clearly presented 4) Figure 3: to which a and b parameters does it correspond? 5) Figure 4: how can I judge by visual inspection that black dots represent the best performing transformation if I do not know about the empirical relationship (figure 4 presents the model?)? 6) Table 1: for sulfate and EC (half of the database) the coefficient of determination for n = 5 (optimal) and n= ïĆě (log-log) is almost the same. What the improvement is? 7) Comparison between observation an model only appear at the very end (figure 6) but no comparison is provided with the log-log transformation (where the improvement is?).

Provided the previous motivation I very much regret to say that in my opinion the paper cannot be accepted for publication.
* * *

---

## Short Comment (SC1) · 4 Sep 2019

Please find below the working link to the response file:

https://www.hydrol-earth-syst-sci-discuss.net/hess-2019-325/hess-2019-325-AC1-supplement.pdf

[Figure]

---

## Author Comment (AC1) · 4 Sep 2019

Dear Colleague,

Thank you for your review. Please find below a detailed response to the points you raised:

- **Although the topic is relevant to the current literature and well within the scope of the journal, I am struggling to understand what the exact nature of the problem is, and how and why the proposed work represents an improvement of the current knowledge about the research topic.**

In this technical note, our purpose was to present as factually as possible a new mathematical representation for the concentration-discharge relationship. We chose the technical note format because we thought it was more adapted to a paper, which only aimed at discussing a mathematical formulation. We hope that the answers brought below will help you better understand our purpose. In any case, we will use your comments to improve our manuscript.

- **The literature review presented in the introduction section (rather rushed) does not at all bring the reader to the idea that a different data transformation, beyond the log-log transformation, is desirable by the scientific community for the representation of Q-C relationship and for what reason it should be.**

Because this is a technical note, we tried to go straight to the point in the introduction; this is probably where the impression of "rushed" comes from. You are perfectly right to mention that there has been (to our knowledge) no direct critic of the log-log transformation in the literature. However, in papers like that of Moatar (2017), the variety of shapes is a clear indication that the log-log transformation lacks generality (we will add this point in the revised manuscript). In a recent paper of the same group, Minaudo et al. (2019) mention that "*fitting a single linear regression on C-Q plots is sometimes questionable due to large dispersion in C-Q plots (even log transformed)*". We also believe that the recent advent of high-frequency time series allows better scrutinizing the C-Q relationships (cf. l. 44-46).

- **The motivation provided in the "About the excess of log-log transformation" section (the change of the shape of the Q-C relation) is quite weak. Authors probably reach an interesting point/motivation when they introduce (line 111) the problem of the representation of high flow discharge data concentration that arises for high frequency database but, surprisingly, when they come to the results, they mention the difficulties of the model to reproduce this type of data (line 151). But many more questions come about the scientific idea.**

Because our new mathematical representation comes to address a problem unsolved by the log-log transformation, we thought that we had to discuss the shortcomings of the log-log transformation. We chose a graphical demonstration (Figure 2) to show that data do not "line up" after transformation. We will make this point clearer in the revised manuscript.

- **Why authors choose the Box-Cox transformation? Aren't there alternative? If they do not compare the performance of the log-log transformation with the proposed two-sided power transformation, how can the reader guess it is an improved representation? How is the improvement demonstrated by the authors?**

We used the Box-Cox transformation (the two-sided power transformation) because (i) it has the requested flexibility and converges towards the classical log-log transformation for high $n$ values, (ii) from our point of view it is the simplest alternative to the one-sided power transformation, (iii) it is almost universally known in the field of statistics and time series analysis. We will make this point clearer in the revised manuscript.

We agree that we did not provide an exhaustive numerical evidence of the superiority of the Box-Cox transformation (we only showed how the coefficient of determination is improved in Table 1). Below, we added in Table 1 a column computing the RMSE of prediction:

**Table 1: Coefficient of determination ($R^2$) and RMSE calculated for n =1 (no transformation), n = optimal value for Box-Cox transformation (Figure 4) and n->∞ (log-log transformation) for each ion and for EC. Note that while the $R^2$ is computed between transformed values, the RMSE is computed between untransformed values.**

| ion | $n$ | $R^2$ | RMSE |
|---|---|---|---|
| Sodium | $n$ = 1 (no transformation) | 0.54 | 1.11 mgL$^{-1}$ |
| | $n$ = 3 (optimal) | 0.73 | 0.97 mgL$^{-1}$ |
| | $n \rightarrow \infty$ (log-log) | 0.53 | 1.22 mgL$^{-1}$ |
| Sulfate | $n$ = 1 (no transformation) | 0.32 | 3.06 SmgL$^{-1}$ |
| | $n$ = 5 (optimal) | 0.81 | 2.00 SmgL$^{-1}$ |
| | $n \rightarrow \infty$ (log-log) | 0.77 | 2.21 SmgL$^{-1}$ |
| Chloride | $n$ = 1 (no transformation) | 0.52 | 3.34 mgL$^{-1}$ |
| | $n$ = 3 (optimal) | 0.88 | 1.92 mgL$^{-1}$ |
| | $n \rightarrow \infty$ (log-log) | 0.69 | 2.91 mgL$^{-1}$ |
| EC | $n$ = 1 (no transformation) | 0.38 | 60.02 µScm$^{-1}$ |
| | $n$ = 5 (optimal) | 0.79 | 37.26 µScm$^{-1}$ |
| | $n \rightarrow \infty$ (log-log) | 0.74 | 41.32 µScm$^{-1}$ |

- **1) The introduction section is quite rushed and presents a figure published elsewhere by other authors. Generally figures are not included in the introduction but if needed why do not use authors own data?**

Our aim in the introduction was to illustrate the fact that the log-log transformation could sometimes be well-adapted, but we found no such case in our own dataset. We understand that this can be misleading for the reader. In the revised paper, we will remove this figure and replace it by a citation.

- **2) Figures frequently do not indicate neither the range of variability of the data not the unit of measurements**

We had removed the unit of measurements and the range of variability in order to focus on the shape of the scatterplot. We understand that it can be misleading for the reader and we will replace them in the revised paper.

- **3) The dataset used for the analysis is not clearly presented**

Because a technical note should be very short, we had kept the dataset description as short as possible. We will extend its description in the revised paper.

- **4) Figure 3: to which $a$ and $b$ parameters does it correspond?**

Figure 3 only shows the data (transformed and untransformed), and $a$ and $b$ refer to the fitted model. Their values will depend on the objective function(s) chosen for fitting; this is why we cannot mention them at this point (they will be given in Table 3).

- **5) Figure 4: how can I judge by visual inspection that black dots represent the best performing transformation if I do not know about the empirical relationship (figure 4 presents the model?)?**

We mention in lines 85-86 that "The optimal shape could be chosen numerically (see Table 1), but we first followed the 86 advice of Box et al. (2016, p. 331) and did it visually". Figure 3 provides a graphical illustration, while Table 1 provides the numerical demonstration.

- **6) Table 1: for sulfate and EC (half of the database) the coefficient of determination for n = 5 (optimal) and n= ï´C ˇe (log-log) is almost the same. What the improvement is?**

See our response above (updated table 1 with RMSE).

- **7) Comparison between observation and model only appear at the very end (figure 6) but no comparison is provided with the log-log transformation (where the improvement is?).**

We now have a numerical comparison (RMSE) in Table 1 and we will modify Figure 6 in the revised paper, showing the comparison between the Box-Cox transformation and the log-log transformation (see below).

[Figure]

**Figure 6: Comparison of observed concentrations with simulated concentrations by: (a) Box-Cox transformation, (b) log-log transformation**

**References**

Minaudo, C. et al., 2019. Seasonal and event-based concentration-discharge relationships to identify catchment controls on nutrient export regimes. Advances in Water Resources, 131: 103379.

Moatar, F., Abbott, B., Minaudo, C., Curie, F., and Pinay, G.: Elemental properties, hydrology, and 228 biology interact to shape concentration-discharge curves for carbon, nutrients, sediment, and major 229 ions, Water Resources Research, 53, 1270-1287, 2017.

---

## Referee Comment (RC2) · Anonymous Referee #2 · 12 Sep 2019

Dear authors and editor,

I saw there were reviewer's comments submitted in Aug 2019, which has been replied by the authors. However, to independently and subjectively review this paper, I did not read their comments and reply on purpose. I am sorry for the potential repeat of the comments with the other referee.

This paper, Technical Note: Improved mathematical representation of concentration-discharge relationships, presents an improved mathematical representation of the empirical relationships between discharge and ions concentrations (C-Q relationship). The core improvement is to modify the log-transformed relationship by the Box and

[Figure]

Cox transformation. Although the topic is of great interest among multi-disciplinary groups, i.e. hydrology, biogeochemistry, agriculture, etc., there exist many pieces of critical confusion and missing information, including the scientific significance and rational, the quality of presentation and the potential impacts on the existing scientific study. Therefore, I do NOT recommend this manuscript to be published in this journal.

Major comments: 1. It seems that the abstract might require more work to gain attentions from potential audiences.

2. It is true that the C-Q relationship has been intensively studied for a very long time. However, the history and existing processes are not thoroughly reviewed in this manuscript. For exampling, in addition to the straightforward power-transformation of C-Q relationship (Eq. 1 in Ln 25), Moarar et al., 2017 proposed a segmented C-Q relationship to detect the change point in long-term C-Q relationships for different ions; Hirsch et al., 2010 and Zhang 2018 utilized Weighted Regressions on Time, Discharge and Season (WRTDS) models to analyze the nutrients export to rivers; Bieroza et al., 2019 evaluated the variations of slopes in C-Q relationships from low-frequency data. It is suggested to provide a more thorough review of current literature and the research gap.

3. There is a missing but critical part in the introduction: the scientific importance and rational of this study. Some questions need to be answered to proceed the manuscript, including: • What is the research gap? • What is the research question in this area? • What is the proposed method or approach to fill the gap and to address the research question? • Why does the proposed method have the potential accordingly? • What are the potential impacts and output that the proposed method will generate?

4. It is very rare to see a figure in the introduction. Please justify the significant importance to include a figure here (Ln 36, Figure 1).

5. It might make more sense if section 2-5 to be re-organized. Some of the descrip-

tion of log-transformed C-Q relationship needs to go to Introduction, and some of the materials need to re-organized as Materials and Methods. A considerable amount of materials needs to go to Results and Discussion.

6. Although this technical note focuses on the development of a method, it is still necessary to briefly introduce the dataset used in this note. It is acceptable to provide a brief summary and a citation to the dataset.

7. With all due respect, the reviewer did NOT see significant differences among n = 3 to n => ∞ in Figure 3 and Figure 4. Please first define what the optimal scenario is and then clearly indicate how to identify the optimal scenario, either visually or numerically.

8. For the multi-objective identification, the authors involved load in the statement. Generally, C-Q relationship only involves the flow (Q) and solute concentration (C). Some researchers also investigated the relationships between Q and load (Basu et al., 2010). Please clarify the definition of C-Q relationship in the manuscript. Please also justify the reason(s) to include load in this relationship.

9. The algorithms of multi-objective identification are not clear. The authors need to provide more narratives to explain why the average of two NSEB would help achieve the optimal goal considering multiple objectives.

10. It seems that the authors considered the two transformation methods as models. For instance, the authors used obs and cal to denote concentration in Eq. 4-8. In addition, the NSE is a widely used indicator to test the performances of numerical models. Please clarify if these transformations are treated as models or not. If so, please discuss the benefits and potential output of numerical modeling of C-Q relationships.

11. There are some potential improvements in the figures. A figure must be able to be interpreted independently without further information in the content or somewhere else. It is disappointing that nearly all the figures fail to meet this requirement. Examples include: • The labels and extent of axis should not be removed (Figure 1-4); • The

units of flow and concentrations are very important (Figure 1-5); • The denotation of symbols needs to be explained. For instance, what is the meaning of [Q] and [C] in Figure 2? The reviewer saw some people utilized [Ca2+] to indicate the concentration of Ca2+. However, the meaning of [Q] and [C] is very confusing.

12. The result section seems somehow weak. If the authors intended to prove that the proposed method performances better, it is required to also present the results for the existing log-transform methods. The editor, reviewers and audience can justify the performances and thus make conclusions. However, in this manuscript, only proposed method is presented.

13. For Figure 6, it is not easy to understand the objective of this figure. If the figure tries to quantify the performance of two-sided power transformation model, there are NO quantitative indicators to show the performances. Additionally, there is no demonstration of the data set for the audience to interpret the results.

14. Due the confusions and missing parts listed above, the conclusions could not be drawn according to the current version of manuscript.

15. It is suggested that the authors should review more publications regarding C-Q relationship.

16. The authors need to make it consistent with the terms used in this manuscript. For example, what are the differences between log-log transformation, power transformation, B-C transformation, logarithm transformation? It is quite difficult to understand so many similar but slightly different terms at different places.

Specific comments: Ln 15: what are the differences between log-log transformation and power transformation?

Ln 23: ". . . give an exhaustive view of the current work on this relationship. . .". Did the authors imply that there is NO research gap given the exhaustive research?

Ln 24: please explain what is "a one-sided power relationship".

Ln 26: generally, no equation is written in the introduction.

Ln 31-33: please discuss the reasons that might cause the differences in clear and unclear relationships for different ions.

Ln 33: please explain why the authors choose to use the data from Neal et al., 2013a and 2013b. Does this phenomenon only appear in their dataset? Or is this phenomenon widely reported by other studies?

Ln 41-42: "For many years, since the size of the C-Q datasets was limited by the cost of chemical analyses, it was difficult to analyze in much detail the precise shape of the C-Q relationship." This statement is NOT correct. By comparing the C-Q relationships between a 3-year high-frequency data and weekly measurements, Duncan et al., 2017 reported that, "...The sensor data corresponding to the 3 years of data overlap (2013– 2015) display essentially the same c‐Q slope as the weekly c‐Q data, even though the number of sensor data points is more than two orders of magnitude greater than the number of weekly points per year. " The reviewer agree with the previous values of high-frequency measurements. However, the value of long-term, low-frequency dataset should also be recognized.

Ln 45: "...all the extremes of the relationship can now be included in the analysis." In this case, please justify how the propose method, B-C transformation, could take the advantage of the appearances of extremes.

Ln 49-50: The statement of "... which provides ..." is irrelevant. Please consider removing it.

Ln 53: please explain what is "RiverLab".

Ln 53: "logarithm transformation" Please make it consistent with "log-log transformation".

Ln 60-62: "...This slightly different shape may be due to the high frequency of the time series (Moatar and Meybeck, 2007) or to catchment dynamics (Kirchner, 2009),..." As

this journal is Hydrology and Earth System Sciences, please discuss further how the hydrological processes, i.e. rainfall, runoff generation, infiltration, might influence the shape of C-Q relationship, for example chloride (Cl-).

Ln 64: please explain "...a continuous alternative...".

Ln 65: please explain the differences between "a progressive alternative" and "a continuous alternative".

Ln 71-73: it is suggested to put this section in Appendix.

Ln 74: please try to avoid using the term "clearly". The audience can be easily confused, as they don't share the same expertise and experiences with the authors.

Ln 79: "...Chloride ions concentrations measured on the Oracle-Orgeval observatory ("RiverLab")..." Repeated information with Ln 53. Please remove it.

Ln 82-83: "To our knowledge, there is no physical or mathematical reason why all ionic species should have a C-Q relationship of the same shape." Actually, there are physical reasons that all ionic species should NOT have a C-Q relationship of the same shape, because the hydro-biogeochemical processes that control the transport and reaction of ions are different. For example, chloride (Cl-) is mostly treated as a non-reactive ion, which indicates that the hydrological processes are the critical factors for C-Q relationship. In contract, nitrate (NO3) is highly soluble and reactive, which means the interactions of all the hydro-biogeochemical processes control the C-Q relationships.

Ln 83: "...and EC (Electrical conductivity)..." should be "... and Electrical Conductivity (EC)..."

Ln 85-86: "...but we first followed the advice of Box et al. (2016, p. 331) and did it visually.." Please clarify how to visually identify the optimal shape.

Ln 86-87: "Figure 4 shows the most adapted power transformation..." Please clarify what a power transformation is.

Ln 97-103: it seems that the optimal shape was identified based on the greatest value of $R^2$. Please include this information in the content and discuss why the greatest $R^2$ will help identify the best shape.

Ln 108-110: it is very confusing. The reviewer has some difficulties to understand this part.

Ln 104-146: The whole multi-objective identification section is not well-organized and the reviewer has some difficulties to understand it. Please consider re-organized it and clearly state the objective of this section.

Ln 150: "..the entire calibration dataset..." If the transformation is treated as a model, please separate the whole dataset independently into calibration and validation sub-dataset. And then report the numerical indicators (i.e. $R^2$) for both calibration and validation sub-dataset.

Ln 150-151: Given the obvious and intensive noise in the scatter-plot for each ion in Figure 6, the statement of "..fit very well.." is quite questionable.

Ln 151-152: "On can only mention..." Please double check the language.

Ln 164-165: "The two-sided power transformation we proposed is a valid and progressive alternative" The current results and discussion cannot support this statement.

Ln 167-168: "The simulated concentrations for the 3 ions and the EC show a good performance." Further evaluation of model performances is required to draw this conclusion.

References

Basu, N. B., Destouni, G., Jawitz, J. W., Thompson, S. E., Loukinova, N. V., Darracq, A., ... & Rao, P. S. C. (2010). Nutrient loads exported from managed catchments reveal emergent biogeochemical stationarity. Geophysical Research Letters, 37(23).

Bieroza, M. Z., Heathwaite, A. L., Bechmann, M., Kyllmar, K., & Jordan, P. (2018). The

concentration-discharge slope as a tool for water quality management. Science of the Total Environment, 630, 738-749.

Duncan, J. M., Welty, C., Kemper, J. T., Groffman, P. M., & Band, L. E. (2017). Dynamics of nitrate concentration–discharge patterns in an urban watershed. Water Resources Research, 53(8), 7349-7365. Hirsch, R. M., Moyer, D. L., & Archfield, S. A. (2010). Weighted regressions on time, discharge, and season (WRTDS), with an application to Chesapeake Bay river inputs 1. JAWRA Journal of the American Water Resources Association, 46(5), 857-880.

Moatar, F., Abbott, B. W., Minaudo, C., Curie, F., & Pinay, G. (2017). Elemental properties, hydrology, and biology interact to shape concentration–discharge curves for carbon, nutrients, sediment, and major ions. Water Resources Research, 53(2), 1270-1287.

Zhang, Q. (2018). Synthesis of nutrient and sediment export patterns in the Chesapeake Bay watershed: Complex and non-stationary concentration-discharge relationships. Science of the Total Environment, 618, 1268-1283.

---

## Author Comment (AC2) · 19 Sep 2019

Dear Colleague,

Thank you for your review, which will help us improve our technical note. Please find below a detailed response to the points you raised:

**There exist many pieces of critical confusion and missing information, including the scientific significance and rational, the quality of presentation and the potential impacts on the existing scientific study.**
1. **It seems that the abstract might require more work to gain attentions from potential audiences.**
2. **It is true that the C-Q relationship has been intensively studied for a very long time. However, the history and existing processes are not thoroughly reviewed in this manuscript. For exampling, in addition to the straightforward power-transformation of C-Q relationship (Eq. 1 in Ln 25), Moarar et al., 2017 proposed a segmented C-Q relationship to detect the change point in long-term C-Q relationships for different ions; Hirsch et al., 2010 and Zhang 2018 utilized Weighted Regressions on Time, Discharge and Season (WRTDS) models to analyze the nutrients export to rivers; Bieroza et al., 2019 evaluated the variations of slopes in C-Q relationships from low-frequency data. It is suggested to provide a more thorough review of current literature and the research gap.**
3. **There is a missing but critical part in the introduction: the scientific importance and rational of this study. Some questions need to be answered to proceed the manuscript, including: ă˘A ´c What is the research gap? ă˘A ´c What is the research question in this area? ă˘A ´c What is the proposed method or approach to fill the gap and to address the research question? ă˘A´c Why does the proposed method have the potential accordingly? ă˘A´c What are the potential impacts and output that the proposed method will generate?**

We wanted to keep the abstract short because this is a technical note. We will expand it.
In the revised paper we will better contextualize the purpose of this technical note. However, we want to make it clear that the only purpose of this note is to show that the log-log transformation should not be considered as the unique approach to link concentration and discharge Naturally, the log-log transformation may be sufficient for some data sets (and this is precisely what we wanted to illustrate by the Figure 1 in the introduction). The article does not aim to oppose two methods, but to show that the log-log transformation is a specific case of the more general Box-Cox transformation. This is why we did not develop the introduction and discuss all the existing research and development on this subject. We referred to rather exhaustive reviews and focused on the new formula, which we did never encounter in the C-Q literature previously.

4. **It is very rare to see a figure in the introduction. Please justify the significant importance to include a figure here (Ln 36, Figure 1).**

Our aim in the introduction was to illustrate the fact that the log-log transformation could be well-adapted for C-Q datasets. We understand that this can be misleading for the reader. In the revised paper, we will remove this figure and replace it by a citation.

5. **It might make more sense if section 2-5 to be re-organized. Some of the description of log-transformed C-Q relationship needs to go to Introduction, and some of the materials need to re-organized as Materials and Methods. A considerable amount of materials needs to go to Results and Discussion.**
6. **Although this technical note focuses on the development of a method, it is still necessary to briefly introduce the dataset used in this note. It is acceptable to provide a brief summary and a citation to the dataset.**

We will check the possible formats of technical notes in HESS and re-organize the sections for more pedagogy and clarity. We had intentionally introduced the dataset very briefly to avoid any discussions other than mathematical. Note also that a long and detailed paper dealing with this dataset has been published recently in the same journal (Floury et al., 2017).

7. **With all due respect, the reviewer did NOT see significant differences among n = 3 to n =>∞in Figure 3 and Figure 4. Please first define what the optimal scenario is and then clearly indicate how to identify the optimal scenario, either visually or numerically**.

We completely disagree on this point: all the colleagues to whom we showed this figures saw the difference. But anyhow, we will add in the revised version the RMSE in Table 1 (see below) in order to show the difference between the efficiency of the different formulas. We will add that our objective is a straight line.

**Table 1. Coefficient of determination ($R^2$) and RMSE calculated for n =1 (no transformation), n = optimal value for two-sided power equation and n $\rightarrow$ ∞ (log-log transformation) for each ion and for EC**

| ion | $n$ | $R^2$ | RMSE |
|---|---|---|---|
| | $n = 1$ (no transformation) | 0.53 | 0.75 mgL$^{-1}$ |
| Sodium | $n = 3$ (optimal) | 0.73 | 0.66 mgL$^{-1}$ |
| | $n \rightarrow \infty$ (log-log) | 0.53 | 0.83 mgL$^{-1}$ |
| | $n = 1$ (no transformation) | 0.32 | 2.11 SmgL$^{-1}$ |
| Sulfate | $n = 5$ (optimal) | 0.81 | 1.38 SmgL$^{-1}$ |
| | $n \rightarrow \infty$ (log-log) | 0.77 | 1.53 SmgL$^{-1}$ |
| | $n = 1$ (no transformation) | 0.52 | 2.30 mgL$^{-1}$ |
| Chloride | $n = 3$ (optimal) | 0.88 | 1.32 mgL$^{-1}$ |
| | $n \rightarrow \infty$ (log-log) | 0.69 | 2.01 mgL$^{-1}$ |
| | $n = 1$ (no transformation) | 0.38 | 56.85 µScm$^{-1}$ |
| EC | $n = 5$ (optimal) | 0.79 | 35.29 µScm$^{-1}$ |
| | $n \rightarrow \infty$ (log-log) | 0.74 | 39.14 µScm$^{-1}$ |

8. **For the multi-objective identification, the authors involved load in the statement. Generally, C-Q relationship only involves the flow (Q) and solute concentration (C). Some researchers also investigated the relationships between Q and load (Basu et al., 2010). Please clarify the definition of C-Q relationship in the manuscript. Please also justify the reason(s) to include load in this relationship.**
9. **The algorithms of multi-objective identification are not clear. The authors need to provide more narratives to explain why the average of two NSEB would help achieve the optimal goal considering multiple objectives.**

We do not use the load at all like Basu et al. (2010): these authors compute annual loads, which they compare to annual discharge, whereas we compute 30 minutes loads. We use loads to give more weight to the extreme low concentrations, load is used as a discharge-weighted concentration (which it is).

Using the "discharge-weighted concentration" (i.e. the load), we avoid the issue of under-representation of high discharge/low concentrations measurement points, which usually prevent the extreme points to be accounted for in the overall model fit.

The two-criterion graph in Fig. 5 ("Pareto plot") shows that we can find a compromise between efficiency in representing the loads and efficiency in representing the concentrations. The "compromise point" is obvious for most of the cases, and the average of the two efficiency criteria is the easiest way to define numerically this point. We will add a sentence on this in the final version of the paper.

10. **It seems that the authors considered the two transformation methods as models. For instance, the authors used obs and cal to denote concentration in Eq. 4-8. In addition, the NSE is a widely used indicator to test the performances of numerical models. Please clarify if these transformations are treated as models or not. If so, please discuss the benefits and potential output of numerical modeling of C-Q relationships.**

The two "formulas" can be considered "models. Fitting a model of the C-Q relationship allows to reconstitute fluxes (i.e. loads) based on discharge only.

11. **There are some potential improvements in the figures. A figure must be able to be interpreted independently without further information in the content or somewhere else. It is disappointing that nearly all the figures fail to meet this requirement. Examples include: ǎA ´c The labels and extent of axis should not be removed (Figure 1-4); ǎA ´c The units of flow and concentrations are very important (Figure 1-5); ǎA ´c The denotation of symbols needs to be explained. For instance, what is the meaning of [Q] and [C] in Figure 2? The reviewer saw some people utilized [Ca2+] to indicate the concentration of Ca2+. However, the meaning of [Q] and [C] is very confusing.**

In the revised paper, we will improve the figures and associated legends.

12. **The result section seems somehow weak. If the authors intended to prove that the proposed method performances better, it is required to also present the results for the existing log-transform methods. The editor, reviewers and audience can justify the performances and thus make conclusions. However, in this manuscript, only proposed method is presented.**
13. **For Figure 6, it is not easy to understand the objective of this figure. If the figure tries to quantify the performance of two-sided power transformation model, there are NO quantitative indicators to show the performances. Additionally, there is no demonstration of the data set for the audience to interpret the results.**
14. **Due the confusions and missing parts listed above, the conclusions could not be drawn according to the current version of manuscript.**

The $R^2$ presented in Table 1 is in itself a goodness-of-fit measure. In the revised paper, we will add RMSE in Table 1. We will also modify Figure 6 (see below) to allow a direct visual comparison of the two models' fits.

[Figure]

Figure 6: Comparison of observed concentrations with simulated concentrations by: (a) two-sided power transformation model (TSPT); (b) log-log transformation model

**15. It is suggested that the authors should review more publications regarding C-Q relationship.**

This is a technical note and not a review paper. To reach the concision objective, we discuss the literature briefly, and refer the readers to one of the recent reviews on the matter.

16. **The authors need to make it consistent with the terms used in this manuscript. For example, what are the differences between log-log transformation, power transformation, B-C transformation, logarithm transformation? It is quite difficult to understand so many similar but slightly different terms at different places.**

In the revised article, we will better define the transformations and be clearer.

**Specific comments:**

**Ln 15: what are the differences between log-log transformation and power transformation?**

We will change the keywords to avoid confusion and replace « Concentration-discharge relationships; log-log transformation; power transformation; multi-objective calibration » by « Concentration-discharge relationships; power law equation, log-log transformation; two-sided-power formula; multi-objective calibration»

**Ln 23: "▷▷▷give an exhaustive view of the current work on this relationship▷▷▷". Did the authors imply that there is NO research gap given the exhaustive research?**

Our aim was only to give credit to colleagues who worked a lot to produce these reviews. We do not mean that there is nothing left to be done. We will change "give an exhaustive view" by "give an almost exhaustive view".

**Ln 24: please explain what "a one-sided power relationship" is.**

The "power law" is a generic term. The two formulas $y=ax^b$ and $y^b=ax^b$ are both power laws. We use the expression "one-sided" and "two-sided" to differentiate them.

**Ln 26: generally, no equation is written in the introduction.**
Generally yes, but there are many exceptions. In a technical note dealing with an equation, we find it logical to present the equation as soon as possible.

**Ln 31-33: please discuss the reasons that might cause the differences in clear and unclear relationships for different ions.**
You are right to underline that the chemodynamic processes/behaviors are extremely complex, and that we do not have the capacity to detail them in this note. We will change "clear" and "not clear" by "good correlation".

**Ln 33: please explain why the authors choose to use the data from Neal et al., 2013a and 2013b. Does this phenomenon only appear in their dataset? Or is this phenomenon widely reported by other studies?**
This phenomenon has been reported by other studies, which authors will cite them in the revised manuscript.

**Ln 41-42: "For many years, since the size of the C-Q datasets was limited by the cost of chemical analyses, it was difficult to analyze in much detail the precise shape of the C-Q relationship." This statement is NOT correct. By comparing the C-Q relationships between a 3-year high-frequency data and weekly measurements, Duncan et al., 2017 reported that, "▷▷▷The sensor data corresponding to the 3 years of data overlap (2013–2015) display essentially the same c–˘RQ slope as the weekly c–˘ARˇQ data, even though the number of sensor data points is more than two orders of magnitude greater than the number of weekly points per year. " The reviewer agree with the previous values of**

**high-frequency measurements. However, the value of long-term, low-frequency dataset should also be recognized.**

We recognize the great value of medium- and low-frequency datasets. We are in charge of an experimental catchment with that kind of historical datasets that are of great value. We will mention it in the revised manuscript.

**Ln 45: "▷▷▷all the extremes of the relationship can now be included in the analysis." In this case, please justify how the propose method, B-C transformation, could take the advantage of the appearances of extremes.**

See the modified figure 6 above.

**Ln 49-50: The statement of "▷▷▷which provides ▷▷▷" is irrelevant. Please consider removing it.**

We do not understand what is wrong with this statement. We will remove "unprecedented".

**Ln 53: please explain what is "RiverLab".**

We will add a short explanation of the River Lab, but it will be not detailed here. Floury et al. (2017) have done it already.

**Ln 53: "logarithm transformation" Please make it consistent with "log-log transformation".**

We will change "logarithm transformation" by "log-log transformation".

**Ln 60-62: "▷▷▷This slightly different shape may be due to the high frequency of the time series (Moatar and Meybeck, 2007) or to catchment dynamics (Kirchner, 2009),▷▷▷" As this journal is Hydrology and Earth System Sciences, please discuss further how the hydrological processes, i.e. rainfall, runoff generation, infiltration, might influence the shape of C-Q relationship, for example chloride (Cl-).**

We will add a short sentence about catchment dynamics.

**Ln 64: please explain "▷▷▷a continuous alternative▷▷▷".**

By "Continuous alternative" we mean that the Box Cox transformation is a continuous generalization of the log log transformation.

**Ln 65: please explain the differences between "a progressive alternative" and "a continuous alternative".**

We will keep "progressive" and remove "continuous"

**Ln 71-73: it is suggested to put this section in Appendix.**

We think that this part of explanation is really important to some readers which may not be familiar with it.

**Ln 74: please try to avoid using the term "clearly". The audience can be easily confused, as they don't share the same expertise and experiences with the authors.**

We will remove "clearly" in the revised manuscript.

**Ln 79: "▷▷▷Chloride ions concentrations measured on the Oracle-Orgeval observatory ("RiverLab")▷▷▷" Repeated information with Ln 53. Please remove it.**

We will remove this repetition in the revised manuscript.

**Ln 82-83: "To our knowledge, there is no physical or mathematical reason why all ionic species should have a C-Q relationship of the same shape." Actually, there are physical reasons that all ionic species should NOT have a C-Q relationship of the same shape, because the hydro-**

biogeochemical processes that control the transport and reaction of ions are different. For example, chloride (Cl-) is mostly treated as a non-reactive ion which indicates that the hydrological processes are the critical factors for C-Q relationship. In contract, nitrate (NO3) is highly soluble and reactive, which means the interactions of all the hydro-biogeochemical processes control the C-Q relationships.

Thank you for bringing this point. We will add a sentence on this topic.

**Ln 83: "▷▷▷and EC (Electrical conductivity)▷▷▷" should be "▷▷▷and Electrical Conductivity (EC)▷▷▷"**

We will change it in the revised manuscript

**Ln 85-86: "▷▷▷but we first followed the advice of Box et al. (2016, p. 331) and did it visually.." Please clarify how to visually identify the optimal shape.**

Of course, you are right; it is easy to do but almost impossible to explain. We will refer to the $R^2$ and RMSE.

**Ln 86-87: "Figure 4 shows the most adapted power transformation▷▷▷" Please clarify what a power transformation is.**

In this case the power transformation is a two-sided-power transformation. It will be changed in the revised manuscript.

**Ln 97-103: it seems that the optimal shape was identified based on the greatest value of R2. Please include this information in the content and discuss why the greatest R2 will help identify the best shape.**

We will specify the $R^2$ and add the RMSE in a new Table 1 (see the first reviewer reply).

**Ln 108-110: it is very confusing. The reviewer has some difficulties to understand this part.**

We will modify this section (see also the answer to the point 8 and 9)

**Ln 104-146: The whole multi-objective identification section is not well-organized and the reviewer has some difficulties to understand it. Please consider re-organized it and, clearly state the objective of this section.**

We will modify this section (see also the answer to the point 8 and 9)

**Ln 150: "..the entire calibration dataset▷ ▷ ▷" If the transformation is treated as a model, please separate the whole dataset independently into calibration and validation subdataset. And then report the numerical indicators (i.e. R2) for both calibration and validation sub-dataset.**

We will include a validation on an independent dataset in the revised manuscript.

**Ln 150-151: Given the obvious and intensive noise in the scatter-plot for each ion in Figure 6, the statement of "..fit very well.." is quite questionable.**

We will be more modest in the revised manuscript.

**Ln 151-152: "On can only mention▷▷▷" Please double check the language.**

OK

**Ln 164-165: "The two-sided power transformation we proposed is a valid and progressive alternative" The current results and discussion cannot support this statement.**

We hope that with the new figure 6 and Table 1 the improvement will be more obvious.

**Ln 167-168: "The simulated concentrations for the 3 ions and the EC show a good performance." Further evaluation of model performances is required to draw this conclusion.**

We hope that with the new figure 6 and Table 1 the improvement will be more obvious.

**References:**

Basu, N. B., Destouni, G., Jawitz, J. W., Thompson, S. E., Loukinova, N. V., Darracq, A., Zanardo, S., Yaeger, M., Sivapalan, M., and Rinaldo, A.: Nutrient loads exported from managed catchments reveal emergent biogeochemical stationarity, Geophysical Research Letters, 37, 2010.

Floury, P., Gaillardet, J., Gayer, E., Bouchez, J., Tallec, G., Ansart, P., Koch, F., Gorge, C., Blanchouin, A., and Roubaty, J. L.: The potamochemical symphony: new progress in the high-frequency acquisition of stream chemical data, Hydrol. Earth Syst. Sci., 21, 6153-6165, 2017.